# Decreasing the Cholesterol Burden in Heterozygous Familial Hypercholesterolemia Children by Dietary Plant Stanol Esters

**DOI:** 10.3390/nu10121842

**Published:** 2018-12-01

**Authors:** Alpo Vuorio, Petri T. Kovanen

**Affiliations:** 1Mehiläinen Airport Health Centre, 01530 Vantaa, Finland; 2Department of Forensic Medicine, University of Helsinki, 00014 Helsinki, Finland; 3Wihuri Research Institute, 00290 Helsinki, Finland; petri.kovanen@wri.fi

**Keywords:** phytosterol, stanol, diet, familial hypercholesterolemia, coronary heart disease, hypercholesterolemia, children, family, atherosclerosis, low-density cholesterol

## Abstract

This review covers the current knowledge about plant stanol esters as a dietary treatment option for heterozygous familial hypercholesterolemia (he-FH) children. The current estimation of the prevalence of he-FH is about one out of 200–250 persons. In this autosomal dominant disease, the concentration of plasma low-density lipoprotein cholesterol (LDL-C) is strongly elevated since birth. Quantitative coronary angiography among he-FH patients has revealed that stenosing atherosclerotic plaques start to develop in he-FH males in their twenties and in he-FH females in their thirties, and that the magnitude of the plaque burden predicts future coronary events. The cumulative exposure of coronary arteries to the lifelong LDL-C elevation can be estimated by calculating the LDL-C burden (LDL-C level × years), and it can also be used to demonstrate the usefulness of dietary stanol ester treatment. Thus, when compared with untreated he-FH patients, the LDL-C burden of using statin from the age of 10 is 15% less, and if he-FH patients starts to use dietary stanol from six years onwards and a combination of statin and dietary stanol from 10 years onwards, the LDL-C burden is 21% less compared to non-treated he-FH patients. We consider dietary stanol treatment of he-FH children as a part of the LDL-C-lowering treatment package as safe and cost-effective, and particularly applicable for the family-centered care of the entire he-FH families.

## 1. Introduction

This review discusses the current knowledge about plant stanol esters as a dietary treatment option for heterozygous familial hypercholesterolemia (he-FH) children. Since the majority of reviews on dietary phytosterols (the common term for plant sterol and plant stanols) focus on plant sterols [1,2], we solely discuss the role of supplementation of the diet with plant stanols. For a comprehensive coverage of dietary supplementation with both plant sterols and stanols, we refer to the recent European Atherosclerosis Society (EAS) position paper in which the topic is discussed comprehensively at the population level [3]. Esterication of plant stanols and plant sterols with fatty acids, with the resultant stanol and sterol esters, has allowed their large-scale incorporation into different food products such as spreads, margarine, and yogurt, and even low-fat foods [4]. Accordingly, dietary consumption of supplemented phytosterols in nutraceutical usually refers to the consumption of their esterified forms.

It is estimated that as many as 4.5 million people in Europe suffer from the heterozygous form of familial hypercholesterolemia [5]. Patients with he-FH have a highly elevated lifelong serum LDL-cholesterol (LDL-C) level, and, in fact, such a difference in LDL-C levels between he-FH and normal subjects already exists in he-FH newborns [6]. The strong pathogenic role of elevated LDL-C level in early atherogenesis is exemplified by the more severe and very rare (estimated prevalence ranging from 1:1,000,000 to 1:300,000) homozygous form of FH (ho-FH), in which the LDL-C level is extremely high and causes the development of coronary and aortic atherosclerosis already during the fetal life [7,8]. In this condition, early diagnosis and the prompt initiation of dietary and highly efficient lipid-lowering therapy are critical [9]. An early dietary and medical treatment is essential also in he-FH because the elevated LDL-C leads to an early onset of atherosclerotic changes in the arterial tree. Indeed, if left untreated, stenotic coronary atherosclerosis has been observed in he-FH males already in their twenties and in he-FH females in their thirties [10]. Especially in males, premature acute myocardial infarction (AMI) may occur already in early adulthood [5,11].

FH is the most common inherited metabolic disease, and it is caused by a mutation in the low-density lipoprotein receptor (*LDLR)* gene [12], apolipoprotein-B gene *(apoB)* [13], or the proprotein convertase subtilisin type 9 *(PCSK9)* gene [14]. Most commonly, the FH-causing mutation is found in the *LDLR* gene, and indeed, so far, more than 1,700 mutations in this gene have been discovered, while much fewer mutations in the *apoB* gene or in the *PCSK9* gene cause FH [12,13,14,15]. The prevalence of he-FH has been historically one case out of 500 persons. However, currently, this prevalence figure has been re-evaluated, and the current estimation of the prevalence of he-FH is about one out of every 200–250 persons [16,17,18,19].

Although a healthy diet is an essential part of the treatment of he-FH children, according to current recommendations, medical treatment in he-FH is also necessary already in childhood starting from the age of 8 to 14 years [20,21,22,23,24,25,26]. Before the statin era, anion exchange resins (cholestyramine and colestipol) were used to lower the LDL-C level [27]. However, resins were poorly tolerated by he-FH children, and therefore, they have been largely abandoned as a treatment choice. Since the introduction of statins (3-hydroxy-3-methyl-glutaryl coenzyme A reductase inhibitors), i.e., since the early 1990s, this class of drugs has been the cornerstone medication for he-FH children, and, fortunately, they have proven generally to be both efficient and safe [28].

## 2. Cholesterol-Lowering Diet Compared with Dietary Interventions to Increase Intake of Plant Stanols

As stated already in the Introduction, the term phytosterols is used to cover both plant sterols and plant stanols. Phytosterols can be extracted from vegetable oils and are also present in nuts, fruits, and vegetables [29,30]. The stanols can be produced by the hydrogenation of plant sterols [31]. In Finland, as an example, the natural dietary intake of phytosterols is about 300 mg/day [32,33]. Plant stanols, again, represent a very low share of the phytosterols; their daily natural intake is only about 20 mg [32,33].

In the Cochrane review including only controlled studies, in which cholesterol-lowering diets were compared with dietary interventions aimed at increasing the intake of plant stanols, only one study was found in which he-FH children were involved [34]. This study was carried among one ho-FH child and 14 he-FH children, and this six-week double-blind crossover study evaluated the effect of rapeseed oil margarine without or with sitostanol esters (3 g/day) [35]. Serum LDL-C decreased significantly (approximately by 15%), when he-FH patients used rapeseed margarine supplemented with sitostanol instead of rapeseed margarine without sitostanol. In the single ho-FH patient who participated in this study, serum LDL-C decreased from 17.7 mmol/L to 16.1 mmol/L. In this careful study, in which the absorption and synthesis of cholesterol were monitored by gas–liquid chromatography, it was found that cholesterol absorption decreased, and cholesterol synthesis was compensatorily increased. The authors concluded that dietary treatment by stanol-enriched rapeseed margarine is effective and safe in pediatric he-FH patients.

In a study of 19 he-FH families in the Finnish North Karelia, dietary treatment with stanol ester margarine was examined [36]. A total of 24 he-FH children aged three to 13 years, with a mean age of 9 ± 1 (SE) years participated in this study, out of which eight were males and all of whom had the same FH North Karelia (FH-NK) mutation. Their levels of serum LDL-C were found to decrease, on average, by 18% during the 12-week study period when the children were treated by rapeseed margarine (2.24 g/day stanols) compared to baseline low-fat and low-cholesterol diet without added stanols (Table 1) [36]. In this study, the serum retinol concentration and the α-tocopherol-to-cholesterol ratio remained unchanged. The he-FH-NK patients with the highest cholesterol absorption efficiencies reduced their adjusted campesterol levels most efficiently, revealing that the dietary stanol treatment is most beneficial among individuals with a high baseline cholesterol absorption rate and a low cholesterol synthesis rate (Figure 1). The findings are similar to those obtained in the above-described earlier study of 14 he-FH children on stanol ester, in which the serum LDL-C levels decreased approximately 18% from the baseline level [35].

Regarding ho-FH children, it should be noted that in the study carried in an FH-family having two heterozygous parents and one homozygous child, dietary consumption of either plant stanol esters or plant sterol esters reduced the levels of serum cholesterol in the heterozygous parents by about 14% and in the homozygous child by about 9% [37]. However, the consumption of sterol esters increased the serum plant sterol concentrations in the ho-FH child 14-fold compared with those in normal or heterozygous subjects, while the consumption of stanol esters decreased them [37]. These data suggest that it would be preferable to use dietary stanols for lowering cholesterol in ho-FH children. This inference may be extended to include also he-FH children, as in both cases, a lifelong treatment lays ahead.

## 3. Stanol Esters and Statins

According to the recent guidelines provided by The Task Force for the Management of Dyslipidaemias of the European Society of Cardiology (ESC) and European Atherosclerosis Society (EAS), functional foods with plant stanols (at least two g/day with the main meal) may be considered in children (from the age of six years) with he-FH [38]. This recommendation is equivalent to the earlier EAS Consensus Panel Review [4].

Only very few studies exist in which hypercholesterolemic patients, and especially he-FH patients, are on a stanol diet on top of statin treatment [39,40]. One of these studies, which was not placebo-controlled, demonstrates the effectiveness of the daily consumption of rapeseed oil margarine (2.24 g/day stanols) on top of 20 mg or 40 mg of simvastatin [36]. In this six-week study, the baseline serum LDL-C level decreased significantly by about 20%, i.e., from 4.09 ± 0.18 mmol/L to 3.27 ± 0.18 mmol/L (SE), and this decrease was independent of simvastatin dose. The effectiveness of a stanol ester diet can be understood when considering the homeostatic control of cholesterol metabolism. Accordingly, by decreasing cholesterol synthesis, statins also cause a compensatory increase in cholesterol absorption [41]. Moreover, the addition of stanol ester as a cholesterol absorption inhibitor on top of the statin reduces the enhanced cholesterol absorption. The effect of stanol ester diet on top statins has been also found among patients with type 1 diabetes [42], type 2 diabetes [43], and among postmenopausal women with coronary heart disease (CHD) [44]. Of note, the plant stanols are practically non-absorbable, while a fraction of the dietary plant sterols is absorbed [45].

Interestingly, statin treatment induces increased plasma plant sterol levels, while plant stanols decrease such statin-induced elevation of circulating plant sterol levels [45]. Accordingly, there has been some concern of potential tissue accumulation with ensuing harmful local effects of plant sterols when they are a supplement to a daily diet in high amounts. To address this concern, the presence and potential harmful effects of plant sterols in stenotic aortic valves removed at valve replacement surgery have been examined, until now, in three human studies [46,47,48]. In the first study by Helske et al., of the total number of 82 patients studied 13 had consumed supplemented plant stanol or plant sterol esters regularly on a daily basis prior to the valve replacement [46]. Aortic valves were collected randomly from a subpopulation of patients (*n* = 21), and of these patients, four had consumed and six had not consumed plant stanol or plant sterol ester supplements regularly (duration not characterized) prior to the surgery, while no information on supplement consumption was available from 11 patients. It appeared that the higher the intestinal absorption of cholesterol, the higher the plan plant sterol contents in the stenotic aortic valves. Moreover, the sterol/cholesterol-ratios in serum closely correlated with those in the valves. However, no differences in plant sterol concentrations in either serum or aortic valves between individuals with (*n* = 4) and without (*n* = 6) using plant stanol or sterol ester supplements were detected. In the second study by Weingärtner et al., of 82 consecutive patients admitted to hospital for elective aortic valve replacement, a structural interview revealed that 10 patients had consumed plant sterol esters either irregularly or regularly for up to 4 years [47]. Patients who had consumed plant sterol esters showed significantly higher plasma concentrations of plant sterols both in serum and the valve tissue. In the third more recent randomized, double-blind controlled intervention study by Simonen et al., 36 patients planned for aortic valve replacement surgery consumed margarine without (*n* = 11) or with plant stanol ester (*n* = 12) or plant sterol ester (*n* = 13) supplementation until valve replacement surgery (2,6 months, on average) [48]. In this study of short duration, it appeared that the consumed plant stanol or sterol esters did not accumulate in the valves, neither did their consumption change the valvular structure, or the numbers of inflammatory cells, such as the macrophages or mast cells, in the aortic valves. Finally, it is important to note that the above-cited studies, which involved adult patients with clinically significant and symptomatic aortic stenosis referred to a hospital of valve replacement surgery, did not include he-FH patients. Therefore, appropriate conclusions of the effects of dietary supplementation with plant stanol or sterol esters on aortic valves in he-FH patients cannot be drawn from these studies, and therefore this important issue remains a challenge for future studies.

Another issue has been the concern related to the adherence to statin therapy if the stanols are used as a part of the diet. Eussen et al. [49] showed that among new statin users the adherence to use statins was not decreased, even when the patients consumed stanol ester supplementation as a part of their daily diet. However, the authors of this study emphasized that the clinicians have the responsibility of reminding the patients that dietary stanol supplementation is an add-on to statin treatment, and certainly not a replacement for statin therapy. Eussen et al. [50] also showed that adding functional foods containing stanols to statin therapy is cost-effective.

## 4. Stanol Esters and Cholesterol Burden

A study by computed tomography angiography has shown that in he-FH patients on non-optimal statin therapy, atherosclerotic plaques started to develop in males and females at a mean age of about 20 and 30 years, respectively [51]. Additionally, the total plaque burden significantly predicted future coronary events.

The cumulative LDL-C burden can be demonstrated by LDL-C burden = LDL-C levels × years of age [52]. We have earlier used LDL-C burden to show the effectiveness of hypolipidemic treatment in he-FH [52,53,54]. LDL-C burden can be used also to demonstrate the usefulness of dietary stanol ester treatment. To this end, we have used the data to show that dietary stanol ester decreases serum LDL-C by approximately −10%. Figure 2 shows three cases of he-FH patients, each of whom was followed up to 18 years of age. In the first case, the patient remained untreated; in the second case, the patient had been on statin treatment since the age of 10 years, and in the third case, the patient had started to consume dietary stanol at the age of six years, and a combination of dietary stanol and a statin at the age of 10 years. Compared with the LDL-C burden of the untreated he-FH patient, the LDL-C burden of the patient on statin was reduced by 15%, and in the he-FH patient on dietary stanol and later on a combination of stanol and statin, the LDL-C burden was reduced by 21%. Thus, the achieved additional decrease by the early initiation of dietary stanol was 6%. However, in this presentation, the synergistic effect of the combination therapy on LDL-C lowering, i.e., the ability of stanol ester to potentiate the LDL-C-lowering effect of a statin was not considered [36].

## 5. Stanol Esters and Family-Centered Treatment Model

There is a saying on the website of the FH Foundation: “We say we never find an individual with FH; we only find families with FH [55]. Such a family-centered approach in FH treatment certainly offers potentially significant advantages. Indeed, the real challenge in the lifelong treatment of he-FH patients is that among patients in general, there is only a weak association between the risk perception and the preventive behavior [56,57]. Therefore, just telling the he-FH child about his/her strongly increased risk of premature CHD and hoping that the child will carefully follow the necessary preventive measures it not sufficient. The he-FH child should also understand and believe that he/she can do something to help reduce the risk, and that an inherited hypercholesterolemia does not cause an immutable risk.

It has been shown that he-FH patients underestimate their risk of suffering from heart disease later on in life and they are also stressed by the uncertainty about the ability of the health behavior changes to reduce their risk of CHD [58]. Thus, while an individual he-FH patient is trying to initiate preventive measures, he/she is also facing the challenge of building his or her personal risk perception. According to a self-regulation model of health and illness, this process is based on both internal and external stimuli [59]. Since in he-FH children, the internal stimuli (physical signs) of the disease are absent, they need an external stimulus to build their risk perception, which in the case of he-FH could be experiencing the illness of a close relative, and thereby obtaining risk information. Such external information can be achieved only in a close collaboration with the members of the own family, or, when necessary, with the members of the extended FH family.

Positive information from family members can be also potentially used to enforce the motivation for CHD prevention. Dietary stanol consumption in the whole family could be this positive non-stressful preventive measure. Furthermore, it is possible to proceed step-by-step with a young he-FH child in a family starting with dietary stanol treatment from six years age toward starting the statin treatment when the child is about 10 years old. The first step is that the whole family, including the non-FH members, starts stanol consumption as a part of their daily eating habits, as has been shown in previous study of FH families [36]. As noted above, the current guidelines advocate stanol treatment for he-FH children from the age of six years onwards [3,38]. It is also noteworthy that starting the consumption of stanol before the initiation of statin treatment has potentially a significant advantage by being a dietary, i.e., a non-pharmacological treatment of the disease. Additionally, starting stanol consumption should be a rather non-stressful change in the health behavior of a child, particularly when the other he-FH members of the family, and ideally also the non-FH members of the family—i.e., the whole family—is consuming stanol.

## 6. Conclusions

This review discusses the current knowledge about plant stanol esters as a dietary treatment option for he-FH children. It is estimated that as many as 4.5 million people in Europe alone suffer from the heterozygous form of familial hypercholesterolemia, and accordingly, the numbers of he-FH children must also be very high. Quantitative computed tomography angiography studies among the he-FH patients have shown that atherosclerotic plaques start to develop in males and females in their twenties and thirties, respectively, and that such early plaque burden significantly predicts premature coronary events. The cumulative LDL-C burden can be calculated by multiplying the LDL-C level by the years of age, and it can be used to demonstrate the usefulness of dietary stanol treatment. We consider dietary stanol consumption as a component of the LDL-C-lowering package to be safe, cost-effective, and applicable not only for an he-FH child, but also for the whole he-FH family. Dietary stanol treatment serves, at its best, as a supportive measure for the early start of a lifelong treatment of the he-FH child.

## Figures and Tables

**Figure 1 nutrients-10-01842-f001:**
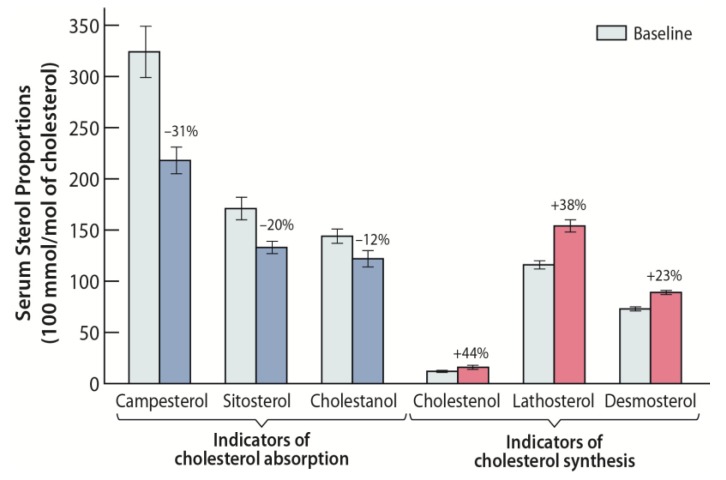
Effect of stanol ester margarine on serum non-cholesterol sterol proportions (100 mmol/mol of cholesterol) among 24 heterozygous familial hypercholesterolemia (he-FH) children aged 3 to 13 years [36]. Stanol ester margarine reduced the ratios of serum campesterol, sitosterol, and cholestanol to cholesterol, reflecting an inhibition of cholesterol absorption. The ratios of serum cholestenol, lathosterol, and desmosterol to cholesterol increased, indicating an increase of cholesterol biosynthesis.

**Figure 2 nutrients-10-01842-f002:**
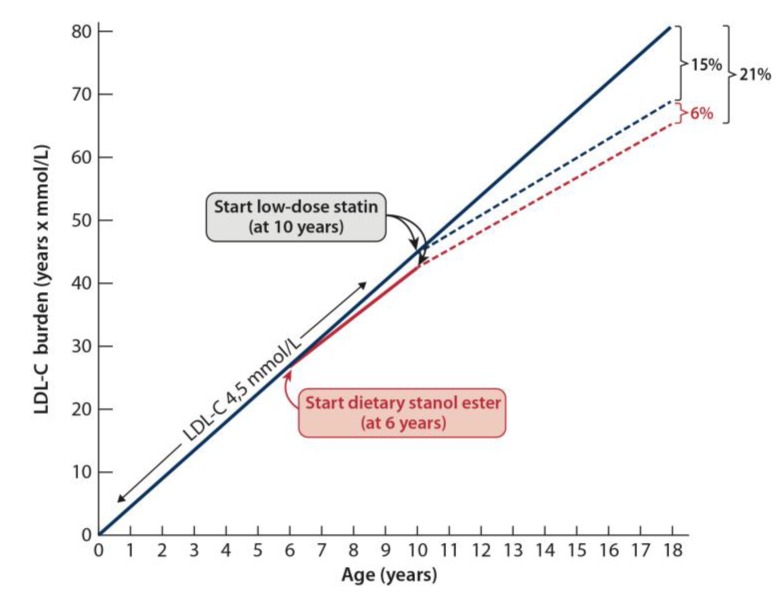
The cumulative low-density lipoprotein cholesterol (LDL-C) burden can be demonstrated by using the calculation = LDL-C levels × years of age [52]. To this end, we have used the data showing that dietary stanol ester decreases serum LDL-C approximately by 10%. Three he-FH patient cases are shown. In the first case, the patient remained untreated, in the second case, the patient had been on statin treatment since the age of 10 years, and in the third case, the patient had started to consume dietary stanol ester at the age of six years and a combination of dietary stanol and a statin at the age of 10 years. Compared with the LDL-C burden of the untreated he-FH patient, the LDL-C burden of the patient on statin was reduced by 15%, and in the he-FH patient on dietary stanol and later on a combination of stanol and statin, the LDL-C burden was reduced by 21%.

**Table 1 nutrients-10-01842-t001:** Effect of dietary stanol ester margarine on serum lipids among 24 heterozygous familial hypercholesterolemia (FH)–North Karelia (FH-NK) children [36].

Lipids	Baseline	Stanol Ester Margarine	Difference, %	*p*
TC	7.41 ± 0.15	6.41 ± 0.15	−13.6 ± 1.9	<0.001
LDL	5.96 ± 0.14	4.94 ± 0.14	−17.9 ± 2.2	<0.001
HDL	1.05 ± 0.05	1.09 ± 0.05	+6.1 ± 3.7	NS
TG	0.90 ± 0.06	0.95 ± 0.07	+5.4 ± 6.7	NS

TC = total cholesterol, LDL = low-density cholesterol, HDL = high density cholesterol, TG = triglycerides, NS = non-significant, lipid values are mmol/L ± SE.

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
