# Peer review of "Decreasing the Cholesterol Burden in Heterozygous Familial Hypercholesterolemia Children by Dietary Plant Stanol Esters"

_nutrients, 2018, doi:10.3390/nu10121842_

Reviewer 1 Report

- In my opinion, the title should highlight the main theme of the review, which is doubtless mainly focused on children. Please, revise the title accordingly, by the inclusion of the word "children".

- In their manuscript, the Authors should also mention the possibility of a combination treatment including phytosterols. For this purpose, I warmly suggest referring to doi: 10.1093/nutrit/nux047 and doi: 10.1186/s12986-017-0214-2, which are of great interest in the field.

Author Response

Please, find attached file.

Reviewer 2 Report

The authors have addressed an important issue. This manuscript is predominantly written clearly and fits the scope of the journal. In this review, the author clearly explained the beneficial effects of plant stanol sterols taken along with statin therapy. This review emphasizes the importance of diet. In this review, the author has explained details of plant stanol sterol clearly in each section in clinical perspective.

·         It would be interesting for readers to submit a more graphical representation of data.

·         Research article published by Anna Keomaki et al. in 2004 (J Lab Clin Med 2004;143:255-62) explained the beneficial role of plant stanol sterols, in their conclusion author raised an important question “ The combination of plant stanol and sterol esters with other hypolipidemic treatments seems to be beneficial in the setting of cholesterol-lowering, but evidence of possible harmful effects of increased plant sterols, especially in FH families during consumption of the plant sterol ester spread, is not available”

What are the authors take on this statement, do author think is there any harmful effects of increased plant sterols

·         The Author has to provide essential characteristics of the study populations- like blood pressure, total cholesterol levels, lipid profiling. Most importantly Author has to provide Campesterol, Beta-Sitosterol and desmosterol levels before and after the start of the diet. In this way, it will be interesting to know how Plant sterols reduce the LDL-Cholesterol.

·         Will there be any short term or long term effects of plant sterols added to a high dose of a statin? What are the authors comment regarding this?

·         In PROCAM study and 4S study of CAD patients, they have reported higher baseline plant sterol concentrations was associated with higher CVD risk. Different studies have suggested an increased risk of atherosclerosis with higher sterol levels, and some studies have reported the opposite, Can the author summaries effect of plant sterols from different clinical studies.

Author Response

Please, find attached file.
